# Highly Effective Modulator Therapy: Implications for the Microbial Landscape in Cystic Fibrosis

**DOI:** 10.3390/ijms252211865

**Published:** 2024-11-05

**Authors:** Kristina N. Valladares, Luke I. Jones, Jarrod W. Barnes, Stefanie Krick

**Affiliations:** 1Gregory Fleming James Cystic Fibrosis Research Center, The University of Alabama at Birmingham, Birmingham, AL 35294, USA; knvallad@uab.edu (K.N.V.); jbarnes@uabmc.edu (J.W.B.); 2Division of Pulmonary, Allergy and Critical Care Medicine, The University of Alabama at Birmingham, Birmingham, AL 35294, USA; jonesli@uab.edu; 3Medical Scientist Training Program, The University of Alabama at Birmingham, Birmingham, AL 35294, USA

**Keywords:** fibrosis, CFTR modulators, airway infections, airway inflammation, aging

## Abstract

Cystic fibrosis (CF) is an autosomal recessive multisystem disorder caused by mutations in the cystic fibrosis conductance regulator (CFTR) anion channel. In the lungs specifically, CFTR mutations lead to changes in mucus viscosity and defective mucociliary clearance. Moreover, people with CF (pwCF) mount an insufficient immune response to invading pathogens, which predisposes individuals to chronic airway disease associated with chronic inflammation, colonization, and recurrent infections by mainly opportunistic pathogens. These chronic infections in the CF lung are typically polymicrobial and frequently harbour multidrug-resistant pathogens, making both treatment and eradication very challenging. During the last decade, the development of highly effective CFTR modulator therapy (HEMT) has led to a breakthrough in treatment options for pwCF. While the majority of pwCF now live longer and have fewer CF exacerbations, colonisation with common respiratory pathogens persists, thereby contributing to chronic inflammation and infection. Interestingly, there are limited reports examining the lung microbiome in the post-modulator era. Since ETI treatment is still quite novel and has only been used for about five years by now, this review will be one of the first discussing the current literature on the effect of ETI on CF pathogens. In addition, we will identify unanswered questions that remain from the effect of HEMT on the CF microbiome.

## 1. Introduction

Cystic fibrosis (CF) is an autosomal-recessive genetic disorder caused by mutations in the cystic fibrosis transmembrane conductance regulator (*CFTR*) gene [1,2]. These mutations lead to a dysfunctional or absent CFTR protein, resulting in the impaired transport of chloride and bicarbonate across the epithelium. Dysfunctional CFTR affects various organs, including the lungs, pancreas, and gastrointestinal tract [3]. In the lungs, defective CFTR leads to hydrational changes at the epithelial surface that result in mucus accumulation, dysfunctional mucociliary clearance, and an altered immune response. These factors collectively contribute to chronic inflammation in the CF airway, leading to progressive epithelial damage and increased susceptibility to upper and lower respiratory infections [3]. Infections in the CF lung are often chronic polymicrobial infections deriving from bacterial, viral, or fungal origins [4,5,6]. The pathogens causing these infections in the CF lung change with age, leading to different pathogens but also the persistence of common pathogens, which become more antibiotic resistant as well as undergo bacterial gene conversions that aid in the avoidance of the host immune system. These adaptions pose new challenges in people with CF (pwCF) in the post-CFTR modulator era, where patients are living longer, warranting the importance of understanding the longitudinal adaptations of pathogens post- highly effective CFTR modulator therapy (HEMT) [4,7,8,9].

CFTR modulator therapies directly target and improve the dysfunctional anion flux across the CF epithelial surface, which ultimately led to a decrease in all-cause mortality in pwCF, including mortality from chronic airway infections. The first CFTR modulator on the market was the CFTR potentiator ivacaftor. Ivacaftor is indicated for pwCF who have the G551D CFTR mutation [10]. Later, a combination of CFTR potentiators and correctors—elexacaftor, tezacaftor, and ivacaftor (Trikafta^TM^, hereby referred to as ETI), was shown to be the most promising CFTR modulator for pwCF, having at least one delta F508-CFTR (del508) mutation in the *CFTR* gene or one other mutation known to be responsive to ETI [10,11]. The del508 mutation is one of the most common CF mutations, and results in the unsuccessful trafficking of CFTR to the plasma membrane [12]. The majority of pwCF treated with ETI have shown a favourable response, including improved sweat testing and lung function, less frequent CF exacerbations, and less sputum production; however, pwCF still experience recurrent respiratory exacerbations and the effects of ETI treatment on infection dynamics is currently not well characterised [13,14,15,16,17]. 

In this review, we will summarise the current literature describing the effects and consequences of CFTR modulation, specifically ETI treatment, on the persistence and virulence of “classical CF pathogens”. Our aim was to screen all studies currently available on PubMed that analyse the effects of HEMT on classical pathogens and the microbial landscape in the CF lung in general and discuss current gaps in knowledge and potential future directions. We believe that research on this topic is timely and warranted since a majority of pwCF currently take CFTR modulators, live longer, and maintain persistent microbial colonisations with recurrent infections that may change the microbial landscape of the CF lung.

## 2. CFTR Modulator Therapy and the CF Microbiome

In non-CF lungs, the respiratory epithelium clears mucus appropriately, which promotes an oxygen-rich environment for colonisation by a diverse repertoire of respiratory microorganisms. Many of these organisms, including *Streptococcus*, *Haemophilus*, *Veillonella*, *Porphymonas*, *Proteus*, and *Prevotella*, can be found throughout the upper and lower respiratory tract and are commensal [18,19]. These commensal colonisers rarely cause disease, as healthy individuals have lung clearance mechanisms such as coughing, efficient mucociliary transport, and a functional innate immune response that leads to a quiescent environment devoid of immune hyper- or hypo-activity. 

In various chronic inflammatory lung diseases, such as CF, thickened mucus and impaired clearance mechanisms caused by CFTR deficiency support an environment that affords pathogenicity to opportunistic organisms, which would otherwise be cleared. In particular, CFTR deficiency leads to dehydrated, nutrient-rich mucus which creates a proper milieu for pathogenicity and colonisation by opportunistic organisms, which leads to a chronic inflammatory environment and susceptibility for infection ultimately leading to airway damage [3,20]. Early in life, the key contributors to respiratory infections in pwCF include *Streptococcus* species and *Haemophilus influenzae* (*Hi*). These common human commensals disseminate into the lung and cause infection, but availability of vaccines and antibiotics significantly lessens the severity of infection, mostly leading to a full recovery. 

As pwCF in the pre-modulator era or those not eligible for ETI grow older, CF lungs exhibit a further lung function decline and are more susceptible to colonisation by opportunistic bacteria that cause recurrent infection and chronic inflammation. These pathogens include but are not limited to *Staphylococcus aureus* (*Sa*), *Stenotrophomonas maltophilia*, *Pseudomonas aeruginosa* (*Pa*), *Achromobacter* species, *Burkholderia cepacia* complex (*Bc*), *Serratia marcescens*, nontuberculous mycobacteria (NTM), and *Aspergillus fumigatus* (*Af*). Most of these pathogens are highly antibiotic resistant, are prone to gene conversion, and many of these bacterial or fungal species can synergise or compete with one another in the chronically inflamed and infected CF lung [9,21,22,23,24].

Because of high instances of multidrug-resistant pathogens in pwCF, advances in treatment developed in recent decades including inhaled and intravenous antibiotics, which led to a decrease in the exacerbation frequency and successful bacterial eradication in some patients [25]. Nevertheless, respiratory failure remains the most common cause of morbidity and mortality in pwCF [25]. In addition to the improvement in lung function, there was substantial hope that the introduction of CFTR modulator therapies would affect clearance/eradication of pathogens in the CF lung.

In 2019, the previously mentioned next-generation combination modulator therapy, ETI was FDA-approved for pwCF, expanding eligibility to 90% of pwCF [11]. Individuals taking ETI experienced a significantly improved FEV_1_ (>60%), less exacerbations, and overall a markedly improved quality of life. In addition, ETI treatment has been demonstrated to have an effect on CF specific pathogens, though results are conflicting [13,16,26]. Some studies demonstrate that the bacterial burden was drastically decreased post-ETI but remained at detectable levels [13,14,26]. Other studies showed that clinical sampling strategies have become challenging as increases in bacterial counts were not due to specific species of CF pathogens, but rather a genus that CF pathogens belong to [16]. *Pa* has been known to be one of the most common pathogens in the adult lung, causing the highest rates of morbidity and mortality amongst pwCF [9,27,28]. However, some studies identified a shift in pathogen dominance months after ETI initiation from *Pa* to *Sa*, more specifically methicillin-resistant *S. aureus* (MRSA), which was isolated from both the airways and nasal cavity [14]. Though these pathogens appear to remain as colonisers, exacerbations still occurred, as the persistence of pathogenic strains remained stable. There are several questions that are still left unanswered, such as whether these pathogens remain commensal colonisers and can they still be isolated from sputum or will pathogenic variants arise in the wake of the post-HEMT era as pwCF live longer, fuller lives [29].

### 2.1. Common CF Pathogens and Their Response to CFTR Modulator Therapies

#### 2.1.1. *Pseudomonas aeruginosa (Pa)*

*Pa* is a Gram-negative, highly antibiotic-resistant bacterium and well known to cause chronic infections in pwCF and associated with increased morbidity and mortality [30]. *Pa* is known to contribute to lung function decline through a large arsenal of virulence factors. In the chronic infectious state, *Pa* can downregulate genes associated with an early virulent mucoid phenotype causing the overproduction of the exopolysaccharide alginate [9,27,30]. In addition, defects in DNA mismatch repair have been seen in the hypermutated strains of *Pa* isolated from pwCF. Small colony variants have also been frequently isolated from pwCF; these strains have enhanced biofilm formation abilities and hyper-piliation [27]. Treatment options for *Pa* infections in pwCF are antibiotics, both used intravenously and inhaled, though prolonged antibiotic treatment contributes to the development of multidrug resistance, making the treatment of CF exacerbations and eradication attempts challenging and often unsuccessful [9,28].

Prior to the FDA approval of ETI, ivacaftor alone was the first modulator indicated for pwCF who had the G115D mutation in at least one copy of the gene, which disrupts the ATP-dependent channel opening and impairing the PKA-dependent activation of CFTR [11,31]. There was an overall decrease in *Pa* positive sputum cultures isolated from pwCF who started ivacaftor treatment, but disease severity could have contributed to the negative conversion of sputum cultures. PwCF who had prior *Pa* infections or a high FEV_1_ (>70%) at the time of treatment exhibited greater chances of sputum culture conversion [32]. While the bacterial burden in pwCF was altered, there was no correlation between improved lung function and decreased *Pa* culture positivity. CFTR correction improves mucociliary clearance, especially in minimally damaged airways, and results in an unfavourable environment for *Pa* to attach and form a biofilm, which may explain in part these phenomena [32]. This could also lead to an improved innate immune response resulting in *Pa* eradication. 

Although ETI has only been available since 2019, studies have shown that the *Pa* bacterial load decreases overall but remains at detectable levels (>10^2^) following the initiation of ETI therapy [33,34,35,36]. Recent studies post-ETI have shown that the colony-forming units of *Pa* in sputum are lowered 10-fold when compared to ivacaftor alone [37]. However, genomic analyses revealed that pwCF remained colonised with the same *Pa* strain pre- and post-ETI treatment. These post-ETI *Pa* strains have the capability to infect other pwCF and experience changes in a vast array of genes, many of which are involved in secondary metabolite synthesis [33]. Some of these genes include genes related to iron acquisition (*pvdQ* and *feoB*), potentially reflecting an ETI-induced adaption or adaption to an improved airway microenvironment. While the capacity for *Pa* transmission remains, the consequences of colonisation with a post-ETI strain are yet to be seen or explored. Other genes needed to produce the siderophores pyoverdine and/or pyochelin were also newly mutated virulence genes post-ETI [33].

These data collectively suggest that the CF lung after ETI treatment remains colonised with a detectable *Pa* bacterial load. These mucoid strains tend to persist and combat treatment better than acute or environmental isolates of *Pa*. One treatment avenue that has been proposed for achieving *Pa* eradication is the inhalation of antibiotics in combination with the ETI treatment [34,38,39]. Results have been inconclusive. Some studies have shown that, when ETI is paired with antibiotics, antimicrobial activity is increased, while with some it is decreased [35]. Post-ETI surveillance studies have shown that while the *Pa* bacterial burden in the lungs decreases, the *Pa* culture burden remains unchanged in the sinus cavity [14]. This could give rise to the potential dissemination to the lung and represent a nidus for chronic sinusitis exacerbations.

#### 2.1.2. *Staphylococcus aureus (Sa)*

*Sa* is a Gram-positive, opportunistic bacterium that can be commonly found on the skin and in the nasal cavity. *Sa* can disseminate into the lower respiratory tract and cause lower respiratory tract colonisation and recurrent infections. This can ultimately lead to progressive lung function decline [40]. *Sa* can be classified into two different groups, methicillin-sensitive *S. aureus* (MSSA) and methicillin-resistant *S. aureus* (MRSA), both of which have been shown to colonise the airways of pwCF [6]. PwCF colonised with MRSA tend to have worsening lung function, predisposing them to increased hospitalisations and significantly affecting their quality of life. These strains have higher transformation rates and genomic plasticity, giving them an adaptive edge when it comes to dealing with the selective pressures in the CF lung environment [24].

Classically, *Sa* is known to be an “early-life” coloniser in pwCF; however, recent evidence suggested that *Sa* colonisation increased in older adults, taking modulator therapies, surpassing *Pa* as the predominant bacterial species [40]. Interestingly, ivacaftor treatment was shown to exhibit the antimicrobial activity against *Sa*, but did not lead to eradication [35]. The combination of ivacaftor together with antibiotics commonly used for *Sa* led to a synergistic effect, which might be useful for future eradication attempts for a suitable CF patient subpopulation [35]. These synergistic effects were observed in multiple clinical isolates, when ivacaftor was paired with linezolid, whereas pairing of ETI with amoxicillin, vancomycin, and teicoplanin, synergistic effects were only seen in a few isolates tested. The additive effects of elexacaftor and ETI were more sporadic or completely absent in the case of linezolid [35].

In addition to ivacaftor, ETI showed the strongest antimicrobial activity by itself, which still seems to be ivacaftor-driven [35]. Elexacaftor alone also negatively affected *Sa* bacterial burden, but not as effective as ivacaftor alone or combined as ETI. Post-ETI surveillance studies showed that *Sa,* specifically MRSA, becomes the dominant organism within the sinus cavity and lungs [14,36]. A more recent study suggested that pwCF had positive cultures for *Sa* for up to 2.5 years post-ETI [37]. Continued epidemiological studies are needed to the characterise long-term effects of ETI on *Sa*.

#### 2.1.3. *Nontuberculous mycobacteria* (NTM)

Although only ~10% of pwCF will develop exacerbations due to infections with NTM, this group of bacterial genera has become of growing concern for pwCF due to its association with progressive lung function decline [41]. These bacteria can be acquired from the environment or through direct transmission between colonised and non-colonised pwCF. There are over 150 identified NTM species, though most NTM species isolated from pwCF are *Mycobacterium avium complex* and *Mycobacterium abscessus* (*Ma*) [41]. NTM is classically multidrug-resistant, and treatment requires the use of a long-term combination regimen that is variable based on the specific strain and the extent of lung disease. In the case of *Ma*, infection with this pathogen is associated with a significant loss of FEV_1_ in pwCF and frequent exacerbations, which are extremely difficult to treat [42]. ETI treatment, however, though led to a reduction in *Ma* sputum culture positivity and eradication occurred as early as one year for more than half of the cohort studied [42]. *Ma* eradication could have been due to CFTR correction, and the improvement in mucociliary clearance with rehydration of the airways, allowing antibiotics to have better “access” to the pathogens and potential reduction in biofilm formation. FEV_1_ measurements only improved in pwCF, who had less NTM burden or when eradication was achieved.

On the other hand, pwCF on ETI treatment, which were not able to clear NTM, still experienced lower FEV_1_ values [42]. In addition, there are several drug interactions between CFTR modulators and different antibiotics used for NTM treatment, which leads to either the exclusion of the use of some of the drugs or necessitates the dosage adjustment of, e.g., ETI [43]. One recent case study reported persisting *Ma* lung disease for 12 years in a pwCF and the initiation of ETI led to *Ma* eradication after 1.5 years [44]. Though this was only one case report, it sheds light on the potential impact that ETI therapy can have on NTM lung infections and warrants further studies and the identification of appropriate CF subpopulations, who might benefit.

#### 2.1.4. *Burkholderia cepacia complex (Bc)*

*Bc* is of particular importance when it comes to chronic infections in pwCF due to its negative impact on prognosis. This opportunistic pathogen is multidrug resistant, causes chronic progressive airway disease, and *Bc* infection is an absolute contraindication for lung transplantation at most centres [45]. While extensive studies have not been performed investigating the effects of ETI on *Bc*, smaller studies have looked at the effect of combination therapies of other CFTR modulators, tezacaftor/ivacaftor, and the CFTR modulator cysteamine along with Seliciclib, known as roscovitine, on *Bc* [46]. The metabolites of roscovitine, known as M3, have been shown to enhance the bactericidal activity of macrophages and correct delF508-CFTR trafficking. Not only was roscovitine able to enhance the macrophage killing of *Bc*, but this was also further enhanced when administered together with the CFTR modulator therapy [46]. Further investigations need to be performed to determine how ETI may impact *Bc* and whether it has the potential to aid in eradication. This would be a “game changer” for pwCF, with advanced lung disease making them potentially eligible for lung transplantation.

#### 2.1.5. *Aspergillus fumigatus (Af)*

While many pulmonary exacerbations in pwCF are caused by viral and bacterial species, there has been a growing interest in *Af*, as many pwCF test positive for this pathogen, linking it with lung function decline [5]. *Af* can affect pwCF in different ways, including the induction of allergic bronchopulmonary aspergillosis (ABPA), which develops due to *Af* sensitisation and an exaggerated immune response leading to characteristic lung changes detected by computed tomography, increased lung inflammation, and reduced lung function [47]. Infections with *Af* that lead to hyphae invasion into bronchial walls is known as invasive pulmonary aspergillosis (IPA). IPA mainly occurs in pwCF, who are severely immunocompromised and/or have developed end stage lung disease. *Aspergillus* bronchitis is more common than IPA, but with *Af* infections underreported overall, the real extent of *Af* affected pwCF is unclear [48]. 

The treatment of *Af*-associated disease is challenging, as antifungal medications have abundant side effects due to their toxicity towards the host. Even with the introduction of ETI, *Af* disease is still common in pwCF and antifungals interact with CFTR modulators, necessitating dose adjustments and still running the risk of adverse effects. Prior to ETI approval, earlier studies suggested that ivacaftor reduced *Af* colonisation [49]. In another study, Currie et al. treated isolated peripheral blood mononuclear cells and polymorphonuclear cells from pwCF with lumacaftor/ivacaftor or ivacaftor alone, demonstrating a significant reduction in *Af*-induced reactive oxygen species [50]. This study also demonstrated that ivacaftor and lumacaftor did not exhibit any antifungal activity [50]. ETI did not exhibit anti-fungal activity against *Af* strains, but reduced *Af* biofilm formation, growth, and altered cell wall properties [51]. Biofilm formation in both the laboratory and clinical isolates of *Af* taken from the sputum of pwCF showed significant decreases in biomass as well as the ability to recover after ETI therapy was reduced. When compared to other modulators, ETI exhibited the strongest effect [51]. Additional studies showed that ETI affects permeability within these *Af*-dominant biofilms, modulating multiple ion channels and potentiating caspofungin at low doses, allowing for cell wall damage within the biofilm [51]. Data also suggested that ETI affected chitin production, which appeared to be strain-dependent, and further studies will need to be performed to examine those findings in detail. Complications involving potential drug interactions between CFTR modulators and antifungals will also need to be considered, as antifungal pharmacokinetics and gastrointestinal absorption may be variable [49]. Antifungal resistance is increasing, which may impact CTFR modulator dosing, so monitoring remains crucial to improve treatment strategies.

#### 2.1.6. Other CF-Related Pathogens

While *Pa* and *Sa* are the dominant bacterial species that colonise and infect the CF lung, there are several opportunistic pathogens found in the CF lung and sputum cultures including *Stenotrophomonas maltophilia*, *Achromobacter* species, *Serratia marcescens*, and *Hi*, along with previously mentioned pathogens [5,6]. While the infection rates of these pathogens vary across the CF population, it has been shown that infection rates are dependent on age and diverse factors; these infections are mostly multidrug resistant and polymicrobial. While in-depth in vitro and in vivo studies assessing the effects of ETI on these pathogens have not been performed, sputum collected from pwCF on ETI therapy showed decreases by 2–3 log10-fold changes in the sputum densities of *Serratia marcescens*, *Achromobacter* species, and *Bc* [16]. Nonclassical CF pathogens, such as *Stenotrophomonas maltophilia*, were not affected by ETI therapy, as many of these bacterial members are commensal bacteria [16]. *Hi*, however, was shown to be increased after the initial ETI treatment, which accounted for several *Haemophilus* species members [16]. Specific polymerase chain reaction experiments were performed on the CF sputum, indicating that *Hi* only accounted for less than 1% of the *Haemophilius* genus sequence reads. However, it does raise the question of whether other *Haemophilus* stains can grow to fill a niche in a “post-ETI lung environment” and might emerge as potential pathogens in the future.

## 3. Discussion

In this review, our aim was to summarise our findings from the reports that address the effects of CFTR modulator therapies, specifically ETI, on the common pathogens found in the CF lung. Most of those studies focused on “classical CF pathogens” including *Pa*, *Sa*, *Bc*, *Af*, and NTM, indicating that ETI or ivacaftor alone could reduce colonisation, biofilm formation, and overall bacterial burden. Numerous clinical studies have concluded that modulator therapy improves the quality of life for individuals by leading to less frequent exacerbations and slowing disease progression, but CF airways remain colonised with common CF pathogens, including the nasal cavity [14,16]. Correcting CFTR dysfunction has led to a reduction in sputum production for many pwCF, but many pathogens persist in damaged airways, undergoing gene mutations and adaptions which confer additional treatment challenges in pwCF [16]. The studies of the pathogen–host responses have been initiated but are only looking at short-term outcomes and focus on certain patient populations and are controversial in some ways. As mentioned beforehand, a lot of the studies performed so far demonstrate the beneficial effects looking at the pathogens themselves, including a reduction in the bacterial burden and a reduction in biofilm, among other pathogen/host outcomes (see Table 1). In the future, it will be of great interest to follow those pwCF in the long term and assess whether those beneficial effects persist with the natural aging process. 

Eradication attempts are currently performed using both antibiotics and ETI, indicating potential synergistic antimicrobial effects [34,38,39], but there is discussion regarding drug interactions between antibiotics and modulator therapy, especially when treating *Af* or NTM. Moreover, there is a need for not only dose adjustments but also more research looking into the pharmacogenetic profiles of individual patients tailoring therapy even more towards a personalised approach [43,52]. One strategy that has been proposed is changing the dosage of ETI based on the antibiotics being described [43]. Another potential strategy that has been explored is developing multifunctional antibiotics that have a direct action on the pathogen, but an indirect effect on the CFTR correction [52]. These explorative treatment strategies help prepare for the unknown long-term usages of modulator therapy.

Genetic analyses for *Pa* and *Sa* have been performed and shown that, post-ETI, many genetic mutations that occur are related to nutrient availability [33,36]. This will also potentially be influenced by the change in body composition of pwCF. Before ETI, malnutrition, especially with advanced lung disease, caused a catabolic state, but ETI initiation led to a significant increase in body weight, changing the nutritional environment and influencing the CF-related morbidities such as CF-related diabetes. It will be very interesting to see in long-term studies how those diverse factors can affect different pathogenic species and their burden in the CF lung. We will have a generation of obese pwCF and might consider the role of newer weight loss medications, such as semaglutide, in their management plans and synergistic or adverse actions with ETI and future CFTR modulator therapies.

We also cannot forget that CF is not cured. Although 90% of pwCF are eligible for CFTR modulator therapy and eligible patients can start ETI early in life, potentially avoiding chronic lung disease development, there are (1) 10% of patients, who are not eligible; (2) patients, who started on HEMT but did not tolerate it; and (3) patients who already have significant CF-related disease and having recurrent infections being colonised with CF-related pathogens. PwCF in those populations will need to be further studied and followed for long-term outcomes and opportunities to participate in gene therapy trials and the effects of those on the CF microbiome.

PwCF on HEMT have a higher life expectancy, which is projected to that of normal physiologic aging. This future population of pwCF may have preexisting pulmonary impairment and potentially also have several other CF-related comorbidities such as diabetes, osteoporosis, and exocrine pancreatic insufficiency. In addition, age-related diseases should also be considered in this population, such as cardiovascular disease, renal disease, and neurodegeneration, which can all lead to the use of polypharmacy and more adverse effects, making “aging pwCF” a novel patient population with unknown obstacles and challenges. Long-term observational studies will be needed to appropriately characterise the effect of ETI and help with personalising and customising treatment options.

In summary, the development and approval of CFTR modulator therapy has substantially changed the patient landscape in cystic fibrosis. Improvements in nutrition and antibiotic treatment options have led to decreases in morbidity and mortality, turning a previously paediatric disease into an “adolescent disease”, but CFTR modulators, especially the most recently approved combination therapy, ETI, has pushed this even further with pwCF now having a chance to experience “physiological aging”. 

## 4. Conclusions

Mortality in pwCF has mostly been linked to lung function decline, which is largely attributed to recurrent infections with various opportunistic pathogens. The majority of those pathogens develop multidrug resistance with time and are difficult to eradicate, leading to a vicious cycle of chronic inflammatory damage. HEMT has revolutionised the treatment of CF-related diseases by increasing both the quality and quantity of life for pwCF; however, only a few studies have assessed the effects of HEMT on the CF microbial landscape. Those studies, however, point towards an overall reduction in bacterial burden and a decrease in virulence factors (Figure 1). In the future, it will be very important to address those questions in the different populations of pwCF, that were early or late to start HEMT, and those who still do not qualify, to understand how the pathomechanisms of chronic inflammation and infection in the CF lung have changed by ETI and to identify additional targets that are promising for future treatment strategies.

## Figures and Tables

**Figure 1 ijms-25-11865-f001:**
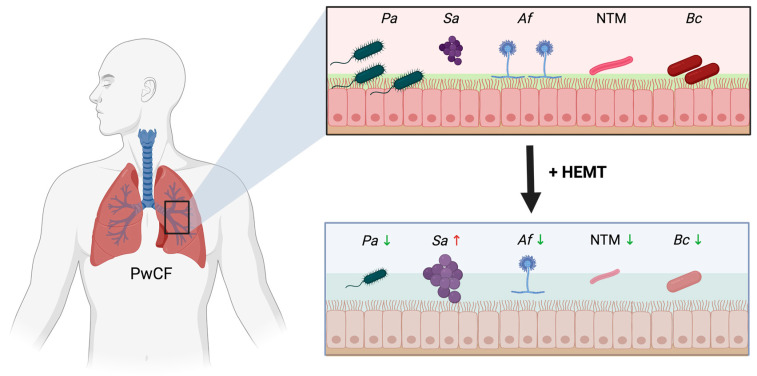
Diagram illustrating the effects of pathogens on the CF lung before and after the introduction of HEMT (created in BioRender. (2024) https://BioRender.com/c71m706, accessed on 21 October 2024).

**Table 1 ijms-25-11865-t001:** Effects of HEMT therapies on different CF-related pathogens.

Pathogen Species	Pathogen/Host Response after HEMT (Reference)
** *Pseudomonas aeruginosa* **	**Decrease/Downregulation:** Attachment [32] Biofilm formation [32] Bacterial burden [14,32,33,34,35,36] Sputum culture positivity [37] **Increase/Upregulation:** Regulation of nutrient availability genes [33]
** *Staphylococcus aureus* **	**Decrease/Downregulation:** Bacterial burden [35] **Increase/Upregulation:** Dominant species [14,36,40] Synergistic effects with antibiotics [35]
** *Aspergillus fumigatus* **	**Decrease/Downregulation:** Colonization [49] Reactive oxygen species [50] Biofilm formation [51] Chitin production [51] **Increase/Upregulation:** Cell wall damage [51] Drug-to-drug interactions [49]
**Nontuberculous mycobacteria**	**Decrease/Downregulation:** Sputum culture positivity [42] **Increase/Upregulation:** Eradication [42,44] Lung Function (FEV_1_) [42] Treatment drug interactions [43]
** *Burkholderia cepacia* **	**Decrease/Downregulation:** Bacterial burden [6] **Increase/Upregulation:** Macrophage-associated killing [46]

## Data Availability

Not applicable.

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
