# Peer review of "Highly Effective Modulator Therapy: Implications for the Microbial Landscape in Cystic Fibrosis"

_ijms, 2024, doi:10.3390/ijms252211865_

Round 1
Reviewer 1 Report
Comments and Suggestions for Authors
The review manuscript “Highly Effective Modulator Therapy: Implications for the Microbial Landscape in Cystic Fibrosis” describes the situation in airway and lung infections and colonization in the era of highly effective modulator therapy for people with cystic fibrosis. The manuscript discusses all the usual bacterial strains in addition to Aspergillus, making it a very thorough and comprehensive review. The authors point out that despite treatment with the highly effective modulators, people with CF still suffer from colonization and infection with the usual culprits. The manuscript deals with an important point and the references are carefully chosen. The conclusions are logical considering the topic discussed and the manuscript is of interest to the field. However, there are some concerns with the manuscript:
1: The authors should discuss the difference between colonization and infection. The one does not necessarily mean the other.
2: The word “microbiome” is mentioned in the abstract, but the manuscript never mentions whole microbiome changes after ETI. Please be clear with what the manuscript is actually about.
3: Use of acronyms: Please go through the text and make sure that you do not use unnecessary acronyms, that all acronyms are explained and that you do not introduce acronyms that are not used.
4: There are some phrasings which need to be revised:
Line 67: The phrase “In non-CF lungs” is repeated twice.
Line 82: There are two problems with this phrase: 1) Please exchange “get” for grow older. 2) People with CF are plural. Please remove the singular indicator “s”.
Line 96: Please rephrase “a lot of” to substantial hope.
Line 102: “ETI treatment has shown to affect CF specific pathogens”. Please rephrase, for example to: ETI treatment has been demonstrated to have effect on CF specific pathogens.
Line 188: What do you mean by broad spectrum scale? Please explain the expression.
Author Response
Dear Editorial Team and Reviewers,
we greatly appreciate the positive response and the constructive feedback for our submitted manuscript. Please see below a point-by-point response to hopefully address all outstanding concerns, which are also addressed in our revised manuscript. We hope that his will make our manuscript suitable for publication.
Yours sincerely,
Stefanie Krick
Reviewer 1:
1: The authors should discuss the difference between colonization and infection. The one does not necessarily mean the other.
Answer: We agree that we did not make this very clear. In the revised manuscript, we describe colonization in the lines 65-72 and further differentiate between colonization and infection in lines 68-84.
2: The word “microbiome” is mentioned in the abstract, but the manuscript never mentions whole microbiome changes after ETI. Please be clear with what the manuscript is actually about.
Answer: We agree that this term does not fit, since our focus was on “classical CF pathogens”, which we used to replace microbiome in the abstract with.
3: Use of acronyms: Please go through the text and make sure that you do not use unnecessary acronyms, that all acronyms are explained and that you do not introduce acronyms that are not used.
Answer: We revised the text accordingly.
4: There are some phrasings which need to be revised:
Line 67: The phrase “In non-CF lungs” is repeated twice.
Answer: Thank you. We deletd the duplicate.
Line 82: There are two problems with this phrase: 1) Please exchange “get” for grow older. Changed 2) People with CF are plural. Please remove the singular indicator “s”. Don’t see this in line 82
Answer: This line is now line 86 in the revised manuscript and we made the appropriate changes.
Line 96: Please rephrase “a lot of” to substantial hope.
Answer: This was changed, now line 100.
Line 102: “ETI treatment has shown to affect CF specific pathogens”. Please rephrase, for example to: ETI treatment has been demonstrated to have effect on CF specific pathogens. Answer: This was changed, now line 105/106.
Line 188: What do you mean by broad spectrum scale? Please explain the expression.
Answer: We apologize for the confusion and revised the sentence now in line 201 accordingly.

Reviewer 2 Report
Comments and Suggestions for Authors
The paper is quite well written. I have some suggestions:
1) Abstract. Interestingly, there are limited reports examining the lung microbiome in the post-modulator era. In this review, we will discuss the current literature and identify unanswered questions that remain from the effect of HEMT on CF lung pathogens and the lung microbiome. Please, underline the novelty of the study.
2) In this review, we attempt to summarize the current literature describing the effects and 58 consequences of CFTR modulation, specifically ETI treatment, on the persistence and 59 virulence of “classical CF pathogens” . We believe that research on this topic is timely and 60 warranted since a majority of pwCF currently take CFTR modulators, live longer, and 61 maintain persistent microbial infections that may change the microbial landscape of the 62 CF lung. I suggest to improve the description of study aim.
3) 2. CFTR modulator therapy and the CF microbiome. I suggest to add a table to summarise the most important topics and references.
4) 3. Discussion 296 The development and approval of CFTR modulator therapy has substantially changed the 297 patient landscape in cystic fibrosis. Improvements in nutrition and antibiotic treatment 298 options have led to decreases in morbidity and mortality turning a previously pediatric 299 disease into an “adolescent disease”, but CFTR modulators, especially the most recent 300 approved combination therapy, ETI, has pushed this even further with pwCF now having 301 a chance to experience “physiological aging”. I suggest to improve the discussion section. It would be beneficial to clearly state the study's objectives. Additionally, presenting the findings more succinctly and providing clarity on the observations, along with a comparison to previous or related research, is essential. Addressing the questions that arise from these findings, supported by existing literature, would also add value to the analysis.
Author Response
Reviewer 2:
The paper is quite well written. I have some suggestions:
1) Abstract. Interestingly, there are limited reports examining the lung microbiome in the post-modulator era. In this review, we will discuss the current literature and identify unanswered questions that remain from the effect of HEMT on CF lung pathogens and the lung microbiome. Please, underline the novelty of the study.
Answer: We apologize we did not make this clearer. We included the novelty of this review in the revised abstract in lines 21-23.
2) In this review, we attempt to summarize the current literature describing the effects and consequences of CFTR modulation, specifically ETI treatment, on the persistence and virulence of “classical CF pathogens” . We believe that research on this topic is timely and warranted since a majority of pwCF currently take CFTR modulators, live longer, and maintain persistent microbial infections that may change the microbial landscape of the CF lung. I suggest to improve the description of study aim.
Answer: We revised our manuscript to make the aim of this review clearer (lines 65-70).
3) 2. CFTR modulator therapy and the CF microbiome. I suggest to add a table to summarise the most important topics and references.
Answer: We added a table, which is not included in the revised manuscript (Table 1: Effects of HEMT therapies on different CF-related pathogens – lines 357-381).
4) 3. Discussion 296 The development and approval of CFTR modulator therapy has substantially changed the 297 patient landscape in cystic fibrosis. Improvements in nutrition and antibiotic treatment 298 options have led to decreases in morbidity and mortality turning a previously pediatric 299 disease into an “adolescent disease”, but CFTR modulators, especially the most recent 300 approved combination therapy, ETI, has pushed this even further with pwCF now having 301 a chance to experience “physiological aging”.
I suggest to improve the discussion section. It would be beneficial to clearly state the study's objectives. Additionally, presenting the findings more succinctly and providing clarity on the observations, along with a comparison to previous or related research, is essential. Addressing the questions that arise from these findings, supported by existing literature, would also add value to the analysis.
Answer: We have revised the discussion section completely. We appreciate the input.
